# Mental Health Consequences of the Three Mile Island, Chernobyl, and Fukushima Nuclear Disasters: A Scoping Review

**DOI:** 10.3390/ijerph18147478

**Published:** 2021-07-13

**Authors:** Misari Oe, Yui Takebayashi, Hideki Sato, Masaharu Maeda

**Affiliations:** 1Department of Neuropsychiatry, School of Medicine, Kurume University, Asahi-machi 67, Kurume 830-0011, Japan; 2Department of Disaster Psychiatry, School of Medicine, Fukushima Medical University, Fukushima 960-1295, Japan; takeb-ky@fmu.ac.jp (Y.T.); hidekis@fmu.ac.jp (H.S.); masagen@fmu.ac.jp (M.M.)

**Keywords:** Three Mile Island disaster, Chernobyl disaster, Fukushima disaster, nuclear power plant accident, mental health

## Abstract

Many individuals who were affected by the Great East Japan earthquake and tsunami and the subsequent Fukushima Daiichi Nuclear Power Plant accident continue to face a challenging recovery. We reviewed the long-term mental health consequences of three major nuclear power plant accidents: the Three Mile Island (TMI, 1979), Chernobyl (1986), and Fukushima (2011) nuclear disasters. We examined the relevant prospective cohort studies and before-and-after studies that covered more than two timepoints, searching four databases (PubMed, Ichushi, PsyArticles, and PTSDPub). We identified a total of 35 studies: TMI, n = 11; Chernobyl, n = 6; and Fukushima, n = 18. The smaller numbers of early-phase studies (within 6 months) of the Chernobyl and Fukushima disasters may also indicate the chaotic situation at those timepoints, as large-scale interviews were conducted in the early phase after the TMI disaster. Although the patterns of effects on mental health outcomes were diverse, more than half of the participants in the studies we evaluated were categorized into low or under-threshold symptom groups in all three disasters. Across the three disasters, the radiation exposure level estimated by the proximity and stigma were the common risk factors for mental health outcomes. Our findings will contribute to a comprehensive understanding of the impact of the worst nuclear accidents in history on the affected individuals’ mental health, and our results illustrate the longitudinal consequences of such disasters.

## 1. Introduction

Severe nuclear disasters have caused serious damage at multiple levels, from the molecular to social and national levels [1]. There is little doubt that the Three Mile Island (TMI) nuclear power plant (NPP) accident, the Chernobyl NPP accident, and the Fukushima Daiichi NPP accident (hereinafter referred to as TMI, Chernobyl and Fukushima) are the three major nuclear disasters of the last half century.

The Three Mile Island (TMI) NPP accident occurred near Harrisburg, Pennsylvania on 28 March 1979. The International Nuclear and Radiological Event Scale (INES) is a tool for communicating the safety significance of nuclear and radiological events to the public [2]. The INES was developed in 1990 by the IAEA and the Nuclear Energy Agency of the Organization for Economic Cooperation and Development (OECD/NEA). Initially, this scale was applied to classify events at NPPs, but later, it was extended and adapted for application to all installations associated with the civil nuclear industry. TMI was categorized as a level 5 “accident with wider consequences” by the INES [2]. With a partial meltdown in the Unit 2 reactor, TMI remains the most serious nuclear disaster to occur in the United States. Although no direct health problems due to the accident’s radiation were reported, the TMI disaster severely damaged the image of nuclear power as a safe source of energy. About 144,000 people within a 24 km radius left their homes for about a week. The major health effect of the accident appears to have been on the mental health of the plant workers and the people living in the region of TMI [3]. The restart of Unit 1, which was conducted 6.5 years after the accident, also had psychological effects on the residents.

The Chernobyl NPP accident occurred on 26 April 1986 in the former Ukrainian Republic of the Soviet Union. More than 600,000 people were registered as emergency and recovery workers (“liquidators”) and some 300,000 residents were relocated [4]. The Chernobyl NPP accident was registered as a level 7 “major accident” by the INES [2]. In this case as well, mental health effects were regarded as the most significant public health consequences across the years [5].

The Fukushima Daiichi NPP accident occurred on 11 March 2011. A huge earthquake with a magnitude of 9.0 struck the Tohoku region in the northern part of Japan. A tsunami following the earthquake damaged all of the cooling systems at the NPP and led to several explosions in the plant buildings. A total of 164,845 residents were evacuated in May 2012. This NPP accident was also registered as a level 7 by the INES [2]. The mental health consequences to survivors have been reviewed [6,7] and high rates of individuals with psychological conditions were reported [6].

In the area of disaster psychiatry, there is a growing interest in studies that investigate the long-term course of psychiatric symptoms over time. The recent development of statistical analysis methods such as latent growth modeling allows researchers to explore long-term trajectories, as well as risk and protective factors related to each trajectory. For example, Norris et al. categorized six types of trajectories in the aftermath of major disasters, i.e., resistance, resilience, recovery, relapsing/remitting, delayed dysfunction, and chronic dysfunction trajectories [8]. Another categorization by Bonanno and Diminich defined six trajectories as follows: continuous, chronic, delayed, recovery, improved, and minimal-impact resilience [9]. Our goal in this review was to extract longitudinal studies of large-scale nuclear disasters throughout history and to examine individual factors associated with long-term outcomes. In addition, we believed it would be particularly useful to provide a broad comparison of the TMI, Chernobyl and Fukushima disasters, so that the similarities and differences among these accidents might be applied to future research.

To the best of our knowledge, there are no reviews that cover the mental health outcomes of residents in the affected areas across these three disasters. In this review, we focused on prospective cohort studies (with non-exposed groups) and before-and-after studies (using only exposed groups) with more than two timepoints, because we have a great interest in the longitudinal trajectories of mental health consequences after a nuclear disaster. Our review’s goals were:(1)To clarify the mental health consequences after the three major NPP accidents (TMI, Chernobyl, and Fukushima) over a long period.(2)To identify positive and negative factors that are associated with the mental health outcomes of people who were exposed to these NPP disasters.(3)To compare the mental health consequences among the three NPP disasters and identify similarities and differences.

## 2. Methods

This scoping review followed the Preferred Reporting Items for Systematic Reviews and Meta-Analysis extension for Scoping Reviews (PRISMA-ScR) statement guidelines. A Protocol is registered in the figshare database (https://doi.org/10.6084/m9.figshare.14113568, accessed on 12 July 2021). Scoping reviews describe both existing literature and as-yet unpublished findings from a range of different study designs and methods [10]. Wider research questions are used in scoping reviews compared to systematic reviews.

We searched the PubMed, Ichushi, PsyArticles, and PTSDPub online databases for publications. PubMed is a free resource supporting the search and retrieval of biomedical and life sciences literature, including MEDLINE. Ichushi is a bibliographic database that was established in 1903 and is updated by the Japan Medical Abstracts Society (JAMAS), a non-profit and non-governmental body. PsyArticles is the database of full-text peer-reviewed studies published by the American Psychological Association and affiliated journals. The PTSDpubs Database (formerly PILOTS) is an extensive post-traumatic stress disorder (PTSD) resource produced by the U.S. Department of Veterans Affairs; it is not limited to the literature on PTSD among veterans. The search formulas for each database are shown in Appendix A.

We included peer-reviewed prospective cohort studies, before-and-after studies, and intervention studies of mental health consequences with two or more timepoints after the TMI, Chernobyl, and Fukushima nuclear disasters. English, German, and Japanese were the target languages. Only studies with 10 or more participants were included in this review. The searches were conducted between 15 July and 31 July 2020.

Two reviewers conducted the study selection: one reviewer screened the records for inclusion, and the other checked the decision. Disagreements were resolved by discussion. Two reviewers conducted the data extraction: one extracted the data, and the other checked the extracted data. Disagreements were resolved by discussion. We used the application Rayyan QCRI [11], which was developed to help expedite the initial screening of abstracts and titles using a process of semi-automation to record the extracted data. Rayyan QCRI allows the user to label each article as to whether or not it is included/excluded and, if it is excluded, for what reason. This makes the extraction of the literature much easier than with ordinary database software.

Figure 1 provides the flow chart of the study selection process. Our database search identified 1693 studies, and after the duplicates were removed, 1611 studies were screened. The first step in the exclusion of ineligible publications was made by evaluating abstracts. The major reasons for exclusion were different themes (i.e., themes on subjects other than NPP accidents), different research areas (e.g., agricultural science), different study designs (e.g., cross-sectional studies), different publication types (e.g., literature reviews), and publication in foreign languages (e.g., Russian). After this process, 116 studies were selected. We then read a full description of these studies and assessed their eligibility as a second step. The most common reason for exclusion was that they had a different study design. In many cases, the abstracts appeared to describe studies with two or more timepoints, but upon reading the full papers, the works actually consisted of two cross-sectional studies conducted on different target groups. Finally, 35 studies were included for this scoping review.

We categorized the 35 studies into categories A, B, and C. Category A was made up of studies on mental health that investigated changes in an identical symptom scale over time. Category B was made up of studies that did not include results of changes of an identical symptom scale over time, as well as studies that used the same results as category A studies. Intervention studies comprised category C.

We evaluated the risk of bias assessments for the studies in categories A and C. The Risk of Bias Assessment tool for Non-randomized Studies (RoBANS) [12] was used for category A studies, and the revised tool to assess the risk of bias in randomized trials (RoB 2) [13] was used for category C studies.

## 3. Results

### 3.1. Characteristics of the Included Studies

In this review, there were 24 category A studies [14,15,16,17,18,19,20,21,22,23,24,25,26,27,28,29,30,31,32,33,34,35,36,37], 10 category B studies [38,39,40,41,42,43,44,45,46,47], and only 1 category C study [48]. The list of included studies is given in Appendix A. The characteristics of all of the studies are summarized in Table 1.

Regarding the presence of a control group, there was a significant difference among three disasters (χ^2^(2) = 16.23, *p* < 0.01). The proportions of studies with a control group were 54.6% for TMI, 100% for Chernobyl, and 11.1% for Fukushima. 

Table 2 provides a summary of the risks of bias in the cohort and before-and-after studies, and Table 3 summarizes the risks of bias in the randomized controlled trial. For three studies (Table 2: Dohrenwend 1981 [15] and Baum 1993 [22] for TMI, and Koscheyev 1993 [25] for Chernobyl), we did not conduct an assessment because the description of the study design was not clear. Among the cohort and before-and-after studies, 9 of 24 studies had a high risk of bias in Selection of Participants, and 19 had a high risk of bias in Blinding of Outcome Assessment. In 17 studies, the risk of bias in Incomplete Outcome Data was unknown. In contrast, the risk of bias in Confounding Variables, Measurement of Exposure, and Selective Outcome Reporting was mostly low. The RCT rated the Overall Risk of Bias as unknown, but the risk of bias was low except for Measurement of the Outcome.

### 3.2. Changes in Mental Health Measures over Time

Among the 24 category A studies, 10 studies reported no changes in the examined mental health measures, 8 reported improvement, 2 reported exacerbation and 4 reported mixed results. There was no significant difference in the patterns of changes in the mental health measures among the three NPP disasters.

#### 3.2.1. TMI

After TMI’s partial meltdown disaster in 1979, the earliest study by Dohrenwend et al. [15] reported an improvement in their subjects’ demoralization score between 1 and 2 months post-disaster, but inconsistent results were observed after those timepoints. There are two specific issues to be considered regarding the TMI disaster; one is the exacerbation of the control group’s mental health symptoms at a mid-term timepoint, i.e., approx. 3–5 years post-disaster, and the other is the restart of the undamaged reactor of TMI at 6.5 years after the accident. Two studies reported an exacerbation of the control group’s mental health symptoms; one study noted that the psychological distress among women at the comparison site where the TMI nuclear reactors are located in western Pennsylvania had increased between 12 and 30 months and between 42 and 54 months after the NPP disaster due to their spouses’ lack of employment following layoffs [19]. The second study demonstrated that at both a comparison site where nuclear reactors are located and another site where coal-fired generating plants are located in western Pennsylvania, there was an increase in psychological distress among mothers between 13 and 30 months and among workers between 31 and 42 months [20].

Two studies indicated the influence of the restart of the undamaged reactor at TMI; one concluded that the distress level of higher-stress-level mothers was exacerbated at the occasion of the restart of the undamaged reactor 81 months post-accident and on the 10th anniversary of the accident [21]. In contrast, a study of residents that compared somatic complaints 1 month before the reactor’s restart and 2 months after the restart revealed that the restart did not heighten stress responses, although the group of subjects directly affected by the TMI disaster reported more somatic complaints at both of these timepoints [23].

#### 3.2.2. Chernobyl

Regarding the Chernobyl NPP disaster in 1986, only four studies [25,26,27,28] were extracted for category A. Two of them were studies of nuclear plant workers (operators or clean-up workers) [25,28], and the other two were investigations of individuals who emigrated to Israel after the accident [26,27]. Only one of the four studies provides data obtained from the former Soviet Union, which collapsed in 1991 [25]. That study was conducted within 2 years after the Chernobyl disaster. The other three studies were conducted more than 6 years after the disaster. These three long-term studies document improvements in depression symptoms, but significant exposure effects remained for PTSD, anxiety, and somatization symptoms [26,27]. The single very-long-term study using Estonia’s Population Registry and Health Insurance Fund database did not detect any significant changes in morbidity for stress reactions, depression, headaches, or sleep disorders [28].

#### 3.2.3. Fukushima

Only one study after the Fukushima NPP disaster was of caregivers [32]; we found three studies of children [35,36] or adolescents (medical university students) [30]. Studies of adult residents [33,34] and employees of the Tokyo Electric Power Company (TEPCO), the operator of the Fukushima NPP [29,31] conducted within 5 years of the disaster (which involved a major earthquake and tsunami, unlike the TMI and Chernobyl accidents) showed improvements of their subjects’ mental health status. A long-term study of residents near the NPP found no changes in their psychological distress and post-traumatic reactions [37].

#### 3.2.4. Temporal Profiles

The temporal profiles of mental health status at more than three timepoints were examined in four studies (TMI, n = 1 [21]; Fukushima, n = 3 [33,34,35]). The temporal profiles were analyzed using a cluster analysis (n = 1) [21], the symptom cluster method (n = 1) [33], and group-based trajectory modeling (n = 2) [34,35], and we identified 2–4 profiles. Across these studies, more than half of the participants were categorized into a low-symptom or under-threshold symptom group.

### 3.3. Risk and Protective Factors According to Psychological Distress

All 35 studies were used for data extraction. A summary of the results of our evaluation of the contributions of risk and protective factors to mental health outcomes is shown in Figure 2. Across the three nuclear disasters, the radiation exposure level estimated by the proximity was the common risk factor for mental health outcomes [15,18,21,27,29,32,35].

Regarding the demographic characteristics of the subjects of the 35 studies, we observed inconsistent results for age. Specifically, a protective effect of older age for participants in the mental health system was reported in a TMI study [14], whereas older age was significantly associated with depression scores among immigrants over the long term after the Chernobyl disaster [26]; older age was also reported as a risk factor after the Fukushima disaster [33,40].

Negative cognitive appraisals of danger, radiation risk, the damaged nuclear plant, and the government’s response were associated with negative mental health outcomes [14,16,21,31,37,46]. In contrast, two behaviors (laughter and physical activity) were revealed as protective factors in studies of Fukushima [33,40,48], including an intervention study of a behavioral activation program [48].

Social support was consistent as a protective factor in the TMI and Fukushima studies. In this review, social stigma was extracted only from studies of the Fukushima disaster.

Regarding the physical condition of individuals affected by the three NPP accidents, several Fukushima studies revealed that not only psychiatric disorders but also physical illness such as diabetes [33,40] were associated with negative outcomes on mental health. The Fukushima Health Management Survey (FHMS) study [49] is a large cohort enrolling all people living in Fukushima Prefecture after the earthquake and consists of four detailed surveys, including a mental health and lifestyle survey [50], whose target population is the residents of evacuation zones. One study of the mental health and lifestyle survey of FHMS demonstrated in a univariate analysis that a medical history of hypertension, diabetes, heart disease, or cerebral vascular disease was associated with the non-recovered pattern [33].

## 4. Discussion

### 4.1. Study Settings

The number of extracted studies differed among the three disasters. The smaller numbers of early-phase studies (within 6 months) of the Chernobyl and Fukushima disasters may also indicate the chaotic situation after those NPP accidents, whereas large-scale interviews were conducted in the early phase after the TMI disaster. In this sense, the post-accident research system of the TMI disaster can serve as an important reference when considering the psychological effects of the accident, even over 40 years after the accident.

The small number of studies of the Chernobyl disaster included in this review is mainly due to the lack of longitudinal studies (e.g., 2–6 years after the disaster). There are several possible reasons for the existence of this or other blank periods. The collapse of the former Soviet Union and our exclusion of publications in the Russian language are likely to be two major reasons. Bromet and Havenaar (2009) described that the breakup of the Soviet Union led to declines in life expectancy and the standard of living and increases in mortality—especially from cardiovascular disease, accidents, and other causes related predominantly to alcohol and smoking [51].

Another reason for a blank period may be that mental health problems caused by the Chernobyl disaster were recognized as “medically unexplained physical symptoms” [4]. The existence of the blank period for mental health issues makes it difficult to reveal the entire picture of the long-term consequences among the survivors of the Chernobyl disaster. Bromet and Havenaar (2009) stated that Western concepts of epidemiology were not widely utilized by investigators in eastern European countries, and they noted that no credible baseline data existed on population health or mental health, suicide, or rates of mental hospitalization after Chernobyl [51].

There is a possibility of misdiagnosing mental health symptoms as brain-organic disorders caused by high dose radiation exposure. For example, a Russian article published six years after the Chernobyl disaster reported the observation of “vegetovascular dystonia,” which was characterized by certain clinical and neurophysiological peculiarities in the form of combined vegetative disturbances and hypochondriac symptoms, signs of the schizoform organic syndrome with diffuse disorders of brain bioelectric activity and irritation of the subcortical structures [52]. Due to the high radiation doses released by the Chernobyl event, radiation-induced brain damage is certainly worthy of consideration. However, it is also likely that some of these cases are actually cases of mental health problems.

In addition, concealment by the Soviet government certainly played a role in the dearth of data following the Chernobyl NPP. Initially, the Soviet authorities tried to conceal the accident from the public [51]. The results of our present analyses revealed considerable differences in the post-accident research system between the TMI and Chernobyl disasters, and the importance of both the (local) government and nationwide research thus emerged. For the Fukushima disaster, the FHMS and the Nuclear Energy Workers’ Support Project (NEWS) were extracted in this review as two large prospective cohorts, and our finding of the low proportions of studies with control groups after the Fukushima disaster may be due to the lack of control group settings of both cohorts. The protocol paper of the FHMS explained that the survey’s primary purpose was providing care to the residents near the Fukushima NPP [49]; therefore, the FHMS was not a random-sampling survey. A complete enumeration inventory survey in the evacuation zone specified by the government was adopted. The fact that the FHMS did not have a control group may be a weak point as a cohort study, but we think that this worked as a strong point for building a post-disaster mental healthcare system. Indeed, brief interventions were provided for the residents near the Fukushima NPP who were identified as being at risk of psychological problems such as depression and post-traumatic stress disorder (PTSD) based on the FHMS, and more than 30,000 affected individuals received telephone counseling over an 8-year period following the accident [53]. In addition, the Fukushima Center for Disaster Mental Health established in 2012 has also been providing outreach services and group interventions for residents in the designated evacuation zone [54]. Synergy effects between surveys and interventions are expected for the field of community mental health. However, random sampling studies with a control group are undoubtedly still necessary.

### 4.2. Changes in Mental Health Outcomes over Time

In this review, we did not find common patterns of changes across the three major nuclear disasters. However, our observation that the more than half of the study participants were categorized into a low-symptom group or an under-threshold symptom group is in line with previous studies. A wide variety of longitudinal trajectories of psychological distress after disasters has been demonstrated [8,55], and our results are also in agreement with this finding. Thus, the mental health trajectories after a nuclear disaster should not be considered a simple decline trend; rather, these are complex, multiple factor-related phenomena.

### 4.3. Risk and Protective Factors

Most of the risk and protective factors that our analyses identified were in agreement with previous cross-sectional studies of the nuclear disasters. We next discuss two factors, the radiation exposure level and “stigma,” as common factors across the three NPP disasters. We also focus on behavioral factors (laughter and physical activity) and physical illness, because they were evaluated in the surveys after the Fukushima disaster.

#### 4.3.1. Radiation Exposure Level

We found that the radiation exposure level was a risk factor for mental health outcomes in all three nuclear disasters. In all the studies, the radiation exposure level was given as the distance from the NPP (proximity). It should be emphasized that proximity does not necessarily correlate with radiation exposure level, and that radiation exposure level is determined not only by external but also by internal exposure. In addition to the actual effects of radiation hazards discussed in Section 4.1., the risk perception of radiation may have played a role in the adverse mental health outcomes. For example, medical university students in the Gomel region, which is located 132 km north of Chernobyl, had anxiety about health and genetic effects due to radiation exposure even 32 years after the accident [56]. In the case of the Fukushima disaster, a study showed that female evacuees who believed that their health was substantially affected by the nuclear disaster were at an increased risk of having poor mental health two years after the disaster [44]. It should also be kept in mind that the risk perception of radiation is not necessarily linked to proximity or actual radiation dose.

#### 4.3.2. Stigma

Although discrimination/slurs were directly extracted as a risk factor of individuals’ mental health status in only the Fukushima studies in our list, social stigma and self-stigma seem to have played important roles in mental health both directly and indirectly. Bromet (2014) reported that people with mental health were stigmatized and marginalized in 1979 (at the time of the TMI disaster), and their examination of the mental health of mothers and their young children was denied access to official birth certificates for sampling purposes by the local government [57]. After the Chernobyl disaster, self-stigma was described in the Chernobyl Forum report as follows: “… individuals in the affected populations were officially categorized as “sufferers”, and came to be known colloquially as “Chernobyl victims,”… This label… had the effect of encouraging individuals to think of themselves fatalistically as invalids… Thus, rather than perceiving themselves as “survivors,” many of those people have come to think of themselves as helpless, weak and lacking control over their future” [4].

Regarding Fukushima, radiation stigma may have had an association with the atomic bomb attacks on Hiroshima and Nagasaki during World War II. A qualitative study of 10 women who were second-generation survivors (SGS) of the atomic bombings in Hiroshima and Nagasaki in 1945 was conducted to shed light on the experience of discrimination and prejudice [58]. No survey respondent of that study had experienced discrimination because they were an SGS. However, one participant answered that her mother and grandmother still hid the fact that her mother was a survivor because they feared prejudice and discrimination. Similar statements have been made among survivors of the Fukushima disaster. Some people mistakenly perceive that women exposed to radiation should not be allowed to marry or reproduce, and thus, many evacuees hide the fact that they were residents of Fukushima after moving to other prefectures [59]. An online survey conducted in 2014 of 750 participants from Hiroshima/Nagasaki, Tokyo, and Fukushima revealed that the perception of radiation stigma of residents in Fukushima was significantly higher than those of the other sites, and the authors also identified a relationship between PTSD symptoms and radiation stigma [60].

#### 4.3.3. Behavioral Factors (Laughter, Physical Activity)

As mentioned above, laughter and physical activity were revealed as protective factors only in studies of the Fukushima disaster. The positive effects of laughter on mental health were reported in a cross-sectional study of the Fukushima disaster [61]. In the FHMS, these two variables (as well as social support) were added in the second survey to search for factors related to resilience after the nuclear disaster. It is noteworthy that the positive effects of laughter on the trajectory of post-traumatic stress symptoms seemed rigid.

Laughter and humor have been known to reduce negative effects during bereavement [62,63]. In a recent intervention study, laughter therapy for cancer patients that combines body exercises and viewing comedy performances had significant positive effects on cognitive function and pain [64]. Although the causality between laughter and mental health should be considered cautiously, community-based interventions based on laughter appear to be beneficial.

The positive effects of physical activity, including regular exercise on depressive symptoms, have been demonstrated in adults and youth [65] and in children and adolescents [66]. In elderly people, frailty and sarcopenia showed an association with depression in meta-analyses [67,68]. Another meta-analysis showed that PTSD is associated with reduced healthy eating and physical activity and increased obesity and smoking [69]. Three studies with a wide range of study participants (children [35], elderly people [40], and mothers [48]) showed positive effects of physical activity on mental health. One of these studies included a behavioral activation program, which is an evidence-based cognitive behavioral therapy that focuses on increasing pleasurable and rewarding activities by using behavioral strategies, including activity scheduling [48]. In that study, the positive effect of just two 90 min sessions was observed at the 1-month follow-up. A program with a longer duration may be effective for the long-term mental health of individuals affected by a nuclear disaster.

#### 4.3.4. Physical Illness

As mentioned above, mental health problems caused by the Chernobyl disaster may have been recognized as medically unexplained physical symptoms, partially because the interplay between physical illness and mental health was not yet fully understood. Poor physical health after traumatic events was suspected [70]; the association between physical illness and mental health outcomes was revealed more recently. For example, an association between metabolic syndrome and depression and anxiety and a relationship between PTSD and diabetes type II [71] and stroke [72] were reported. In the present review, a history of diabetes was identified as a risk factor in two studies [33,40]. Cross-sectional studies conducted after Fukushima also showed associations between metabolic syndrome and mental health outcomes [73]. The significance of comprehensive assessments of concurrent physical and mental health may increase in the future.

### 4.4. Comparison between Nuclear and Natural Disasters

In our study, a diverse range of mental health outcomes was observed. A systematic review on PTSD following disasters demonstrated that some studies following exposure to both natural (e.g., bush fires, floods) and technological (e.g., airplane crashes, explosions) disasters showed reductions in PTSD over time, whereas other studies showed an increase in PTSD prevalence over time [74]. The authors of this review speculated that the number of lives lost may affect the course of the mental health consequences. However, in the case of a nuclear disaster, even if the number of direct fatalities is small, the impact on the mental health of residents would be significant. For example, the number of disaster-related suicides in Fukushima prefecture was much higher than the rates in other prefectures sustaining greater damage from the tsunami (but not severely affected by the nuclear accident) three years after the Great East Japan Earthquake [75]. In this sense, the fact that the number of deaths is not in direct correlation with the magnitude of the effects may be at least partly attributable to the impact of mental health in radiation disasters.

### 4.5. Similarities and Differences

As described in the Introduction section, one of the purposes of this study was to compare the mental health consequences among the three NPP disasters and identify similarities and differences. Based on the results of this study, we can now discuss these similarities and differences. The common features among all three disasters were that the mental health effects are long term, the effects may not be directly related to the number of deaths, and radiation exposure level and stigma are thought to be risk factors. As to the differences, we found considerable variation in post-accident research and mental health support systems among the three accidents (as discussed in Section 4.1.), but we were not able to identify any major differences in the mental health consequences themselves.

### 4.6. Limitations

Our review has several limitations to be considered. Our reference search was limited to four databases (Pubmed, Ichushi, PsyArticles, PTSDpub), and our search formula may not have included all appropriate mental health outcomes. We did not include non-published studies, such as conference abstracts. We did not conduct an independent data extraction by different research groups; however, we alternatively asked a second researcher to cross-check all of the data extraction and resolved disagreements about the results.

The results of the assessment of risk of bias showed that many of the studies included in the scoping review contained items with a high risk of bias. In most studies on mental health in nuclear disasters, the study subjects cannot be randomly assigned, and the outcomes are measured by self-administered scales. Therefore, the high risk of bias in each included study is a limitation of this research area as a whole.

### 4.7. Future Directions

Among the three nuclear disasters that we reviewed, cohort studies are still being conducted for the Chernobyl and Fukushima disasters. In general, it is expected that many cohort studies and before-and-after studies will be continued or newly conducted in the long term. In addition, intervention studies with follow ups will be needed at both the individual and community level.

The target populations of these studies will be the high-risk groups that require mental health and psychosocial support during radiological and nuclear emergencies [76] (see Table 4). Especially for Fukushima, in light of our present findings, studies of subcontracted workers other than TEPCO employees and investigations that include a control group are especially encouraged. It is also necessary to conduct studies from historical and cultural perspectives for the TMI and Chernobyl disasters. For example, it is expected that our understanding of the time period for which no Chernobyl data are currently available and our knowledge of the mental health of TMI workers will eventually grow.

## 5. Conclusions

In this scoping review using prospective cohort studies and before-and-after studies that covered more than two timepoints, long-term mental health consequences and related factors after three major nuclear disasters were presented. We hope that this scoping review will contribute to a comprehensive understanding of the impacts of NPP accidents on mental health in the short and long terms.

## Figures and Tables

**Figure 1 ijerph-18-07478-f001:**
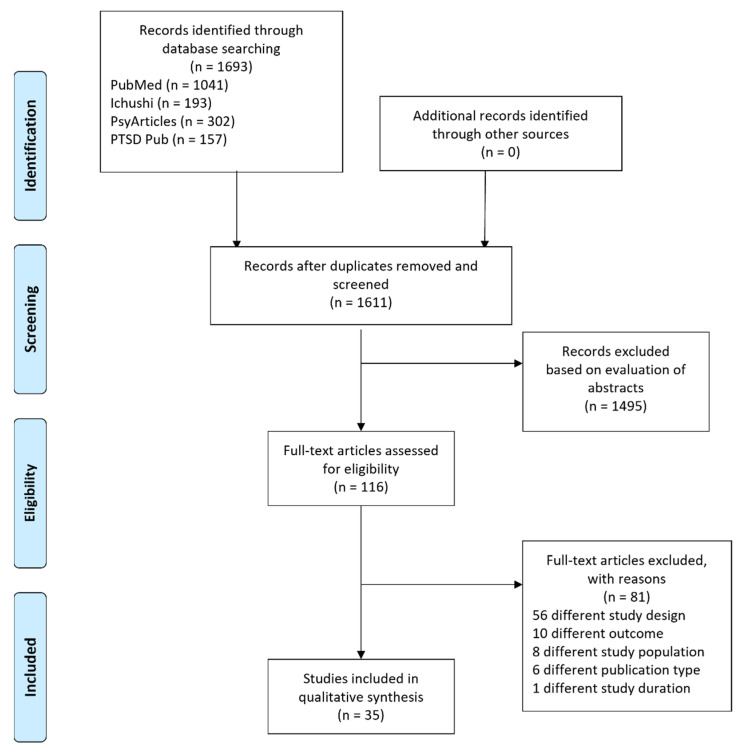
Flow diagram of study selection.

**Figure 2 ijerph-18-07478-f002:**
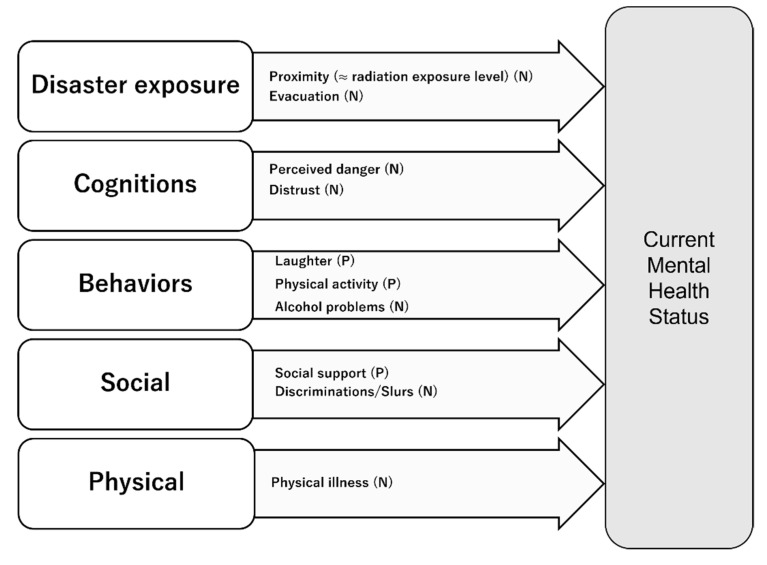
The contributions of risk factors and protective factors to mental health outcomes. P: positive association, N: negative association.

**Table 1 ijerph-18-07478-t001:** Characteristics of the included studies (n = 35).

Disasters	TMI (n = 11), Chernobyl (n = 6), Fukushima (n = 18)
Study participants (multiple answer allowed)	Residents (n = 20), workers (n = 8), mothers or caregivers with/without their children (n = 8), mental health system clients or patients (n = 2)
No. of surveys	Two (n = 19), three (n = 9), four (n = 5), six (n = 1), eight (n = 1)
No. of target populations	Lower than 50 (n = 3), between 50 and 999 (n = 21), more than 1000 (n = 11)
Presence of control group	Yes (n = 14), no (n = 21)
Mental health measure (in Category A, multiple answer allowed)	SCL-90 or SCL-90R (n = 9), IES-R or IES (n =4), K6 (n = 3), PCL-S or PCL-S6 (n = 2) Demoralization score (n = 2), CES-D (n = 2), original distress scale (n = 1), MMPI (n = 1), three-digit ICD-10 code (n = 1), JPSS (n = 1), AIS (n = 1), GHQ-12 (n = 1), emotional symptoms (n = 1), peer relationship (n = 1), victimization (n = 1)

AIS: Athens Insomnia Scale, CES-D: Center for Epidemiologic Studies Depression Scale, GHQ: General Health Questionnaire, ICD-10: International Classification of Diseases, Tenth Revision, IES-R: Impact of Event Scale—Revised, JPSS: Japanese version of the Perceived Stress Scale, K6: 6-item version of the Kessler Psychological Distress Scale, MMPI: Minnesota Multiphasic Personality Inventory, PCL-S: PTSD Checklist—Specific, PCL-S6: 6-item abbreviated version of PCL-S, SCL-90R: Symptom Checklist 90 Revised. TMI: The Three Mile Island.

**Table 2 ijerph-18-07478-t002:** Risk of bias in cohort and before-and-after studies.

Included Studies	Study Design	Selection of Participants	Confounding Variables	Measurement of Exposure	Blinding of Outcome Assessment	Incomplete Outcome Data	Selective Outcome Reporting
**TMI**							
Bromet 1982 [14]	Cohort study	2	1	1	2	3	1
Dohrenwend 1981 [15]	−	−	−	−	−	−	−
Goldsteen 1989 [16]	Before–after study	1	1	1	2	3	1
Goldsteen 1982 [17]	Before–after study	1	1	1	2	3	1
Bromet 1982 [18]	Cohort study	2	1	1	2	3	1
Dew 1987 [19]	Cohort study	2	1	1	2	3	1
Bromet 1990 [20]	Cohort study	2	1	1	2	3	1
Dew 1993 [21]	Before–after study	1	1	1	2	1	1
Baum 1993 [22]	−	−	−	−	−	−	−
Davidson 1991 [23]	Cohort study	2	3	1	1	3	1
Prince-Embury 1995 [24]	Before–after study	1	1	1	2	3	1
**Chernobyl**							
Koscheyev 1993 [25]	−	−	−	−	−	−	−
Cwikel 1998 [26]	Cohort study	2	1	1	2	3	1
Cwikel 1997 [27]	Cohort study	2	1	1	2	3	2
Rahu 2014 [28]	Cohort study	2	1	1	1	1	1
**Fukushima**							
Ikeda 2019 [29]	Before–after study	1	1	1	2	3	1
Kato 2017 [30]	Before–after study	1	1	1	2	1	1
Ikeda 2017 [31]	Before–after study	1	1	1	2	3	1
Sawa 2013 [32]	Cohort study	2	1	1	2	1	1
Oe 2017 [33]	Before–after study	1	1	1	2	3	1
Oe 2016 [34]	Before–after study	1	1	1	2	3	1
Oe 2018 [35]	Before–after study	1	1	1	2	3	1
Oe 2019 [36]	Before–after study	1	1	1	2	3	1
Fukasawa 2020 [37]	Before–after study	1	1	1	2	3	1

The studies are shown in chronological order after each event. We used the Risk of Bias Assessment tool for Non-randomized Studies (RoBANS; Kim et al., 2013) to assess the risk of bias of the included studies. 1 = low risk of bias, 2 = high risk of bias, 3 = unclear risk of bias.

**Table 3 ijerph-18-07478-t003:** Risk of bias in the single randomized controlled trial (RCT) article of Imamura 2016 [48].

Randomization Process	Deviations from the Intended Interventions	Missing Outcome Data	Measurement of the Outcome	Selection of the Reported Result	Overall Risk of Bias
1	1	1	3	1	3

We used the Revised Cochrane risk-of-bias tool for randomized trials (RoB 2; Sterne et al., 2019) to assess the risk of bias of the included study. 1 = low risk of bias, 2 = high risk of bias; 3 = unclear risk of bias.

**Table 4 ijerph-18-07478-t004:** Potential target populations for future studies (revised from “A Framework for Mental Health and Psychosocial Support in Radiological and Nuclear Emergencies,” WHO, 2020).

Target Populations	Reasons
People in close proximity to extremely stressful events, such as an explosion at an accident site	High-dose radiation exposure, death threats
Parents and future parents concerned about the long-term effects of radiation and health of their children	Risks of thyroid cancer, stomach cancer and solid cancer
Children from affected areas	May face discrimination, stigmatization and bullying at school
People with additional physical health needs, such as ill, older or disabled individuals	High risk of health hazards at the time of evacuation
People with a low level of literacy	May struggle to follow advice and instructions provided by risk communicators
First responders, health workers, clean-up workers, reporters and other responders working under hazardous or stressful conditions	Risk of high-dose radiation exposure, burdensome workload
People in residential facilities/institutions (assisted living, retirement homes, correctional facilities)	May not receive enough information, high risk of health hazards at the time of evacuation
Evacuees, as well as the members of hosting communities, whose lives were affected by the evacuation	Drastic changes in living environment
People with pre-existing mental health and psychosocial needs	High risk of worsening symptoms
Workers (and their families) at the nuclear facility where the accident took place	Risk of high-dose radiation exposure, burdensome workload, discrimination/slurs from the public

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
