# Peer review of "Mental Health Consequences of the Three Mile Island, Chernobyl, and Fukushima Nuclear Disasters: A Scoping Review"

_ijerph, 2021, doi:10.3390/ijerph18147478_

Round 1

Reviewer 1 Report

This is a generally well-written, well-design review article. There are some minor issues that need to be addressed: - The authors mentioned using Rayyan QCRI to expedite the initial screening. I would like to see more details (1-2 sentences would suffice) on how the semi-automatic process works. For example, what decisions or screening criteria were made when screening data using the tool? This will enhance reproducibility of the research. - The authors mentioned that 1611 studies were screened. I would like to see detailed numbers of the screening factors of those 1611 studies. For example, xx% were screened out due to [exclusion criteria 1] and xx% were screened out due to [exclusion criteria 2]. Given that the exclusion number is huge (1611/1693!), the authors should explain clearly why such a large number of studies were excluded. - Similarly, among the 116 studies remained, only 35 were included. Why? What were the criteria leading to the exclusion of these 80 studies? (I see the detailed breakdown now in Figure 1, but it would be nice to include in the main text and also include the percentage of the 1611 being excluded in both Figure 1 and the main text.) - In Figure 1, why were records excluded n = 1318? 1611-116 would be 1495? typo? - Table 2 caption is confusing. What does it mean by "risk of bias of in the cohort.."? typo? - I appreciated that the authors conducted the risk of bias assessment. However, the authors only described that they conducted the assessment and the huge Table 2 and table 3 showing the ratings for each study. I would like to see more details in the results section. Particularly, when describing Table 2 and 3 in the main text, please also discuss a brief conclusion of the bias (e.g., how many studies are considered highly bias, how many were ok?). - Also, did the authors factored in the bias risk when they discussed the results? For example, many studies did not show change in mental health status - how many were highly biased among these studies and thus should not be taken too seriously? In short, I think it will help if the authors discussed their results with bias risk in mind.

Author Response

Responses to Reviewer 1

Thank you very much for reviewing this manuscript. Your valuable advice helped us to revise the Methods section in particular.

Comment 1: The authors mentioned using Rayyan QCRI to expedite the initial screening. I would like to see more details (1-2 sentences would suffice) on how the semi-automatic process works. For example, what decisions or screening criteria were made when screening data using the tool? This will enhance reproducibility of the research.

Response: Thank you for your comment regarding the use of Rayyan QCRI. We added a description in the Methods section.

Comment 2: The authors mentioned that 1611 studies were screened. I would like to see detailed numbers of the screening factors of those 1611 studies. For example, xx% were screened out due to [exclusion criteria 1] and xx% were screened out due to [exclusion criteria 2]. Given that the exclusion number is huge (1611/1693!), the authors should explain clearly why such a large number of studies were excluded. - Similarly, among the 116 studies remained, only 35 were included. Why? What were the criteria leading to the exclusion of these 80 studies? (I see the detailed breakdown now in Figure 1, but it would be nice to include in the main text and also include the percentage of the 1611 being excluded in both Figure 1 and the main text.)

Response: Thank you for your comment on the screening process. We agree that further explanation of the inclusion process is needed. Because the decision to exclude a study was made for many different reasons, and many studies were excluded for two or more reasons, it is very difficult provide the percentages for each (for example, articles in Russian were excluded regardless of their contents; however, this does not mean that there are other reasons for exclusion). We therefore found it impossible to draw a Venn diagram of the set. Instead, to account for our process, we added a list of the reasons for exclusion with examples. Although there was also more than one reason for exclusion in the second step, the main reason for exclusion was much clearer than the first step. Therefore, we described the main reason for exclusion in the Figure 1.

Comment 3: In Figure 1, why were records excluded n = 1318? 1611-116 would be 1495? typo?

Response: Thank you for pointing this out. This was a typo and we corrected it.

Comment 4: Table 2 caption is confusing. What does it mean by "risk of bias of in the cohort.."? typo?

Response: Thank you for catching this. This should read “risk of bias in cohort and before-and-after studies”. We corrected it.

Comment 5: I appreciated that the authors conducted the risk of bias assessment. However, the authors only described that they conducted the assessment and the huge Table 2 and table 3 showing the ratings for each study. I would like to see more details in the results section. Particularly, when describing Table 2 and 3 in the main text, please also discuss a brief conclusion of the bias (e.g., how many studies are considered highly bias, how many were ok?). - Also, did the authors factored in the bias risk when they discussed the results? For example, many studies did not show change in mental health status - how many were highly biased among these studies and thus should not be taken too seriously? In short, I think it will help if the authors discussed their results with bias risk in mind.

Response: Thank you for your comment on the risk of bias assessment. We added a description in Section 3.1. We also added an appropriate description in the limitations section (Section 4.6).

Reviewer 2 Report

This manuscript is a scoping review about the mental health (MH) studies of three major nuclear disasters—Three Mile Island, Chernobyl, and Fukushima. Nuclear disasters rarely happen, and this study organizes important lessons learned from these events. I have several comments that might help improve this paper.

Major comments

  1. The authors emphasize in their Abstract that the radiation exposure level is among the major risk factors for MH outcomes. The authors briefly report this statement in Section 3.3 but not without further explanations. The authors also did not include this topic in Figure 2. Furthermore, the authors did not discuss this relationship in the Discussions. I recommend the authors to mention more about this topic in the corresponding sections.
  2. Abstract: The authors describe that “The smaller numbers of early-phase studies (<6 months) of the Chernobyl and Fukushima disasters may also indicate the chaotic situation at those timepoints”. I do agree that post-disaster situations were chaotic for both of these disasters. As for Chernobyl, however, we also need to think about the political situation of the former USSR communism, where secrecy was the norm. Section 4.1 should also be revised accordingly.
  3. The authors also discuss that this blank period in Chernobyl were caused by people recognizing MH problems as medically unexplained physical symptoms (Section 4.1). That is true more or less, but should we also think about a possibility of people attributing MH symptoms as brain-organic disorders caused by high dose radiation exposure? Along with Comment #1, I recommend the authors to discuss this topic with further details.
  4. The authors reported the diversity of MH outcomes in the Results but not in the Discussion. Given the wide-range MH impact of nuclear disasters compared with “conventional” disasters (e.g., floods, hurricanes), it might be worthwhile to tackle this issue.
  5. Discussion, Future directions and conclusion, lines 340-350: The authors are from Japan, so I am wondering if they are speaking about directions for Fukushima studies or nuclear disaster studies in general (e.g. “We expect that many cohort studies and before-and-after studies will be continued or newly conducted”, “Intervention studies at both individual and community levels with follow-ups are also needed”). Could you be explicit about your target population? Also, could we also learn from TMI studies (and not only Chernobyl) from “historical and cultural perspectives”?

Minor comments

  1. Abstract, “the radiation exposure level and stigma seemed to be the common risk factors…”: I recommend the authors to be objective and change “seemed to be” to “were”.
  2. Introduction, lines 30-44: Correct me if I am wrong—levels of nuclear accidents are defined by the INES, which was created by the IAEA and OECD/NEA. Please define INES before explaining “level-7”.
    https://www.iaea.org/resources/databases/international-nuclear-and-radiological-event-scale
  3. Introduction, lines 53-54: Please define “nuclear power plant (NPP)” in its first text appearance.
  4. Methods, lines 90-91: Please defines categories A, B, and C in this section (and not in the Results—the readers have to go back and forth in the manuscript to understand this sentence).
  5. Results, lines 211: Please briefly explain the Fukushima Health Management Survey for readers who are not familiar with this study.
  6. Discussion, line 278: I did not understand the following sentence: “mental health was stigmatized and marginalized in 1979”. Do you mean that compared with 2010s, people had higher stigma to mental health disorders and psychiatric patients were marginalized when the TMI disaster took place?
  7. Discussion, lines 288-289: Could you briefly describe the following statement, with emphasis on hibakusha? “Regarding Fukushima, radiation stigma may have had an association with the atomic bomb attacks on Hiroshima and Nagasaki during World War II.”

Author Response

Responses to Reviewer 2

Thank you for reviewing our manuscript. Your suggestions have helped us to deepen our thinking, especially in regard to the Discussion section.

Major Comment 1: The authors emphasize in their Abstract that the radiation exposure level is among the major risk factors for MH outcomes. The authors briefly report this statement in Section 3.3 but not without further explanations. The authors also did not include this topic in Figure 2. Furthermore, the authors did not discuss this relationship in the Discussions. I recommend the authors to mention more about this topic in the corresponding sections.

Response: Thank you for your comment. We completely agree. We added a new paragraph on the radiation exposure level (Section 4.3.1). In the previous version of Figure 2, we thought that the label "proximity" would be understood as synonymous with the exposure level. However, we see that this is not necessarily the case. We therefore changed “proximity” to “Proximity (radiation exposure level) (N)” in the Figure 2.

Major Comment 2: Abstract: The authors describe that “The smaller numbers of early-phase studies (<6 months) of the Chernobyl and Fukushima disasters may also indicate the chaotic situation at those timepoints”. I do agree that post-disaster situations were chaotic for both of these disasters. As for Chernobyl, however, we also need to think about the political situation of the former USSR communism, where secrecy was the norm. Section 4.1 should also be revised accordingly.

Response: Thank you for your comment. We added a passage on the government concealment to the Discussion (Section 4.1).

Major Comment 3: The authors also discuss that this blank period in Chernobyl were caused by people recognizing MH problems as medically unexplained physical symptoms (Section 4.1). That is true more or less, but should we also think about a possibility of people attributing MH symptoms as brain-organic disorders caused by high dose radiation exposure? Along with Comment #1, I recommend the authors to discuss this topic with further details.

Response: Thank you for your comment. We completely agree. We added a brief passage on brain-organic disorders to the Discussion (Section 4.1).

Major Comment 4: The authors reported the diversity of MH outcomes in the Results but not in the Discussion. Given the wide-range MH impact of nuclear disasters compared with “conventional” disasters (e.g., floods, hurricanes), it might be worthwhile to tackle this issue.

Response: Thank you for your comment. We agree that it would be worthwhile to compare nuclear disasters and natural disasters. We added a subsection entitled “Comparison between nuclear and natural disasters” to the Discussion (Section 4.4).

Major Comment 5: Discussion, Future directions and conclusion, lines 340-350: The authors are from Japan, so I am wondering if they are speaking about directions for Fukushima studies or nuclear disaster studies in general (e.g. “We expect that many cohort studies and before-and-after studies will be continued or newly conducted”, “Intervention studies at both individual and community levels with follow-ups are also needed”). Could you be explicit about your target population? Also, could we also learn from TMI studies (and not only Chernobyl) from “historical and cultural perspectives”?

Response: Thank you for your comment. According to your advice and that of another reviewer, we re-constructed the passage into a subsection entitled “Future directions” (Section 4.7). We separated the treatments of general disasters and specific disaster(s). In addition, target populations were described in detail using a newly added table (Table 4). Finally, we added description about the historical and cultural perspectives of TMI.

Minor Comment 1: Abstract, “the radiation exposure level and stigma seemed to be the common risk factors…”: I recommend the authors to be objective and change “seemed to be” to “were”.

Response: Thank you for pointing this out. We corrected the verb as suggested.

Minor Comment 2: Introduction, lines 30-44: Correct me if I am wrong—levels of nuclear accidents are defined by the INES, which was created by the IAEA and OECD/NEA. Please define INES before explaining “level-7”.
https://www.iaea.org/resources/databases/international-nuclear-and-radiological-event-scale

Response: Thank you for your comment. We agree with your point and revised the second paragraph of the Introduction section accordingly.

Minor Comment 3: Introduction, lines 53-54: Please define “nuclear power plant (NPP)” in its first text appearance.

Response: Thank you for your comment. We added a definition for NPP.

Minor Comment 4: Methods, lines 90-91: Please defines categories A, B, and C in this section (and not in the Results—the readers have to go back and forth in the manuscript to understand this sentence).

Response: Thank you for your comment. We moved the sentences from the Results to the Methods section.

Minor Comment 5: Results, lines 211: Please briefly explain the Fukushima Health Management Survey for readers who are not familiar with this study.

Response: Thank you for your comment. We added a description of the FHMS in the last paragraph of the Results (Section 3.3).

Minor Comment 6: Discussion, line 278: I did not understand the following sentence: “mental health was stigmatized and marginalized in 1979”. Do you mean that compared with 2010s, people had higher stigma to mental health disorders and psychiatric patients were marginalized when the TMI disaster took place?

Response: Thank you for your comment. We corrected the sentence to: “Bromet (2014) reported that people with mental health were stigmatized and marginalized in 1979 (at the time of the TMI disaster)”.

Minor Comment 7: Discussion, lines 288-289: Could you briefly describe the following statement, with emphasis on hibakusha? “Regarding Fukushima, radiation stigma may have had an association with the atomic bomb attacks on Hiroshima and Nagasaki during World War II.”

Response: Thank you for your comment. We completely agree. We added a description on the discrimination of second-generation survivors of atomic bombings in Hiroshima and Nagasaki in the second paragraph of the section entitled “Stigma” (Section 4.3.2).

Reviewer 3 Report

The article deals with an interesting issue regarding the mental health consequences of the major nuclear power plant accidents using the scoping review. However, the following changes are necessary in my opinion:

  • The introduction section is too short and it is necessary to convey the information of the research being conducted by the authors. For example, why the authors are interested in the mental health outcomes of residents in the disaster sites? Were there any prior studies on mental health? Although the authors present the prior studies in the supplemental materials, the introduction section needs to be expanded.
  • Although the Three Mile Island (TMI, 1979), Chernobyl (1986), and Fukushima (2011) nuclear disasters are famous, it might be a good idea to add more description of the three major nuclear power plant accidents separately. 
  • This research needs to enforce its methodology section. Although the authors explain the flow of their research, they need to explain their main research methods, “the scoping review”. What is the difference between scoping review method and the systematic review method?
  • The subtitle “4.5. Future directions and conclusion” is not appropriate. Please make a section for the conclusion. The conclusion section might be separated from the discussion section.
  • This study mentions three goals of the research. I think the third (To compare the mental health consequences among the three NPP disasters and identify similarities and differences) need to be organized more explicitly from the current one.

I also have minor comments that may improve the article:

Line 72 and 77 – “The search formula for the online databases is provided in Supplementary Material 1.” are being duplicated. Please delete one of them.

Line 116 – “Table 2 provides a sum- 115 “a summary of the risks of bias in the cohort and before-and-after studies)” There are unnecessary parenthesis at the end of the sentence. Please delete it

Line 118 – 199321  for Chernobyl), the parenthesis for citation number should be corrected

Line 165 – “≥7 years” Please express it in words, not symbols in the sentence.

Line 234 – Bromet and Havenaar (2009) need to be in line with the citation style of the journal

Throughout the article, the period of the sentences need be placed after the parenthesis of the quotation number. eg) “and national levels. [1]" -> "and national levels [1]."  Please see the instructions for the authors.

Author Response

Responses to Reviewer 3

Thank you for reviewing our manuscript. Your comments were very helpful, especially for helping us to clarify the overall structure of the paper.

Comment 1: The introduction section is too short and it is necessary to convey the information of the research being conducted by the authors. For example, why the authors are interested in the mental health outcomes of residents in the disaster sites? Were there any prior studies on mental health? Although the authors present the prior studies in the supplemental materials, the introduction section needs to be expanded.

Response: Thank you for your comment. We agree with you. We added a new paragraph about our initial motivation for writing this review. We also tried to expand the Introduction section. We did not present the details of mental health issues but have cited a review for each so that you can refer to it if you are interested. In order to emphasize the fact that our study has more than one time point, we would like to refrain from presenting the detailed findings of the cross-sectional studies in the Introduction.

Comment 2: Although the Three Mile Island (TMI, 1979), Chernobyl (1986), and Fukushima (2011) nuclear disasters are famous, it might be a good idea to add more description of the three major nuclear power plant accidents separately.

Response: Thank you for your comment. Again, we completely agree. We revised the text to explain each disaster separately.

Comment 3: This research needs to enforce its methodology section. Although the authors explain the flow of their research, they need to explain their main research methods, “the scoping review”. What is the difference between scoping review method and the systematic review method?

Response: Thank you for your comment. We added an explanation of the scoping review in the first paragraph of the Methods section.

Comment 4: The subtitle “4.5. Future directions and conclusion” is not appropriate. Please make a section for the conclusion. The conclusion section might be separated from the discussion section.

Response: Thank you for your comment. We separated the future directions and conclusion into their own sections.

Comment 5: This study mentions three goals of the research. I think the third (To compare the mental health consequences among the three NPP disasters and identify similarities and differences) need to be organized more explicitly from the current one.

Response: Thank you for your comment. We agree completely. To render the organization of this material more explicit, we added a new subsection, “Similarities and differences” (Section 4.5), to the Discussion.

Comment 6: Line 72 and 77 – “The search formula for the online databases is provided in Supplementary Material 1.” are being duplicated. Please delete one of them.

Response: Thank you for pointing this out. We deleted the second formula.

Comment 7: Line 116 – “Table 2 provides a sum- 115 “a summary of the risks of bias in the cohort and before-and-after studies)” There are unnecessary parenthesis at the end of the sentence. Please delete it

Response: Thank you for pointing this out. We deleted the parenthesis.

Comment 8: Line 118 – 199321  for Chernobyl), the parenthesis for citation number should be corrected

Response: Thank you for pointing this out. We corrected the citation style (we did not submit in this format, because IJERPH allows authors to use the free format option).

Comment 9: Line 165 – “≥7 years” Please express it in words, not symbols in the sentence.

Response: Thank you for pointing this out. We have reduced the use of symbols in the text as much as possible, including in similar passages.

Comment 10: Line 234 – Bromet and Havenaar (2009) need to be in line with the citation style of the journal

Response: Thank you for pointing this out. We corrected the citation of this paper.

Comment 11: Throughout the article, the period of the sentences need be placed after the parenthesis of the quotation number. eg) “and national levels. [1]" -> "and national levels [1]."  Please see the instructions for the authors.

Response: Thank you for your comment. Because we submitted using the free format option, the editorial office has converted the formatting. We corrected this type of inconsistency in the revised version.

Round 2

Reviewer 2 Report

Thank you very much for taking the comments of another reviewer and myself for consideration. While I feel that the manuscript is in a better shape now, I am concerned about the following section:

>>Major Comment 1: The authors emphasize in their Abstract that the radiation exposure level is among the major risk factors for MH outcomes. The authors briefly report this statement in Section 3.3 but not without further explanations. The authors also did not include this topic in Figure 2. Furthermore, the authors did not discuss this relationship in the Discussions. I recommend the authors to mention more about this topic in the corresponding sections.

> Response: Thank you for your comment. We completely agree. We added a new paragraph on the radiation exposure level (Section 4.3.1). In the previous version of Figure 2, we thought that the label "proximity" would be understood as synonymous with the exposure level. However, we see that this is not necessarily the case. We therefore changed “proximity” to “Proximity (radiation exposure level) (N)” in the Figure 2.

Thank you for agreeing that proximity from NPP is not synonymous with radiation exposure level. I also want to emphasize proximity is not necessarily correlated with radiation exposure level—the latter is determined not only by external but also internal exposure. What’s more, the people are more or less affected by their perceived radiation threat, which is not always in line with proximity or radiation dose. I recommend the authors to differentiate perceived threat, proximity, and radiation dose in Figure 2, and edit their Abstract, Results, and Discussion accordingly.

Author Response

Thank you for your comment. We agree with your opinion that proximity is not necessarily correlated with radiation exposure level. Because there are no studies in this paper that directly measure and compare radiation levels, we changed our expression of “radiation exposure level” into “radiation exposure level by the proximity” in the Abstract, the Result section. Figure 2 was changed into “Proximity (≈ radiation exposure level)”. In addition, we added the description regarding your comment in the Section 4.3.1. as “It should be emphasized that proximity does not necessarily correlate with radiation exposure level, and that radiation exposure level is determined not only by external but also by internal.”

The perceived radiation threat has already been described in the paper as “perceived danger”, which is a category of cognition in Figure 2. In order to emphasis of the differentiation, we added a sentence of “It should also be kept in mind that the risk perception of radiation is not necessarily linked to proximity or actual radiation dose.” In the Section 4.3.1.

Reviewer 3 Report

The revised paper properly reflects my comments and has improved a lot after the revision. At this stage I rate it as suitable for publication.

Author Response

Thank you for your positive feedback. We really appreciate it.